The impact of stress on financial decision-making varies as a function of depression and anxiety symptoms

Robinson Oliver J. oliver.j.robinson@gmail.com
Bond Rebecca L.
Roiser Jonathan P.
Institute of Cognitive Neuroscience, University College London , UK
Brambilla Paolo
Electronic publication date: 2015 Feb 12
Publication date: 2015
Volume: 3
Electronic Location ID: e770
Received 2014 Dec 3; Accepted 2015 Jan 26
Copyright: © 2015 Robinson et al.
Copyright year: 2015
Copyright holder: Robinson et al.
License: This is an open access article distributed under the terms of the Creative Commons Attribution License, which permits unrestricted use, distribution, reproduction and adaptation in any medium and for any purpose provided that it is properly attributed. For attribution, the original author(s), title, publication source (PeerJ) and either DOI or URL of the article must be cited.
License URL: https://creativecommons.org/licenses/by/4.0/

Keywords: Stress, Anxiety, Depression, Iowa Gambling Task, Resilience, Threat of shock, Risk-seeking, Harm-avoidance, Vulnerablity

Funding: Medical Research Council Career Development Award MR/K024280/1 This work was funded by an individual Medical Research Council Career Development Award to Oliver J. Robinson (MR/K024280/1). The funders had no role in study design, data collection and analysis, decision to publish, or preparation of the manuscript.

==============================
Stress can precipitate the onset of mood and anxiety disorders. This may occur, at least in part, via a modulatory effect of stress on decision-making. Some individuals are, however, more resilient to the effects of stress than others. The mechanisms underlying such vulnerability differences are nevertheless unknown. In this study we attempted to begin quantifying individual differences in vulnerability by exploring the effect of experimentally induced stress on decision-making. The threat of unpredictable shock was used to induce stress in healthy volunteers (N = 47) using a within-subjects, within-session design, and its impact on a financial decision-making task (the Iowa Gambling Task) was assessed alongside anxious and depressive symptomatology. As expected, participants learned to select advantageous decks and avoid disadvantageous decks. Importantly, we found that stress provoked a pattern of harm-avoidant behaviour (decreased selection of disadvantageous decks) in individuals with low levels of trait anxiety. By contrast, individuals with high trait anxiety demonstrated the opposite pattern: stress-induced risk-seeking (increased selection of disadvantageous decks). These contrasting influences of stress depending on mood and anxiety symptoms might provide insight into vulnerability to common mental illness. In particular, we speculate that those who adopt a more harm-avoidant strategy may be better able to regulate their exposure to further environmental stress, reducing their susceptibility to mood and anxiety disorders.

Introduction

It is well established that stress can precipitate mood and anxiety disorders (de Kloet, Joels & Holsboer, 2005). However, it is also recognised that there exist great differences amongst individuals; some individuals are resilient to stress, whilst others are particularly vulnerable (Kendler, Kuhn & Prescott, 2004). However, the neural and behavioural mechanisms underlying such individual differences in vulnerability remain poorly understood. One potential mechanism by which stress might contribute to mood disorder vulnerability is via its modulatory impacts on behaviour (Dias-Ferreira et al., 2009; Robinson et al., 2013). In this study we sought to explore how the impact of stress on behaviour might vary as a function of individual differences in mood and anxiety symptoms.

We explored the impact of stress on performance of a well-validated financial decision-making paradigm: the Iowa Gambling Task (Bechara et al., 1994). In this task, healthy individuals learn to play from ‘advantageous’ decks of cards—which result in net gains across repeated selections—and to avoid ‘disadvantageous’ decks, which result in net losses (Maia & McClelland, 2004). To induce stress during this task we used the well-validated (Robinson et al., 2013; Grillon, 2008; Davis, 2006; Davis et al., 2010) within-subject, within-session, threat of unpredictable shock paradigm, which has been shown to increase reliably the psychological (Robinson et al., 2013), physiological (e.g., startle responding Schmitz & Grillon, 2012; Grillon, 2008; Grillon et al., 1991) and neural (Mechias, Etkin & Kalisch, 2010; Vytal et al., 2014; Davis, 2006; Robinson et al., 2012a) concomitants of anxiety, in both humans and experimental animals (Davis, 2006; Davis et al., 2010). To explore individual differences in stress reactivity, we recruited a sample of individuals self-identifying as healthy but reporting a range of trait anxiety and depression scores.

A small amount of prior work has explored the influence of stress on this task. Social stress—induced by threat of unprepared public speaking on the Trier social stress test—has been reported to reverse the adaptive bias towards advantageous decks, triggering increased selection of disadvantageous decks (Preston et al., 2007; Van den Bos, Harteveld & Stoop, 2009) (albeit only in some individuals in a complex interaction with individual differences). Other work has shown that individuals at risk of anxiety disorders (Miu, Heilman & Houser, 2008), depressed individuals (Must et al., 2013) and individuals with chronic pain (Walteros et al., 2011) also show an attenuation of this adaptive bias due to increased selection of disadvantageous decks. However, others reported reduced selection of disadvantageous decks in individuals with high trait anxiety (Mueller et al., 2010; Werner, Duschek & Schandry, 2009). What might underlie these discrepancies is unclear. Given the critical interplay between trait vulnerability and environmental stress in stress-diathesis models of mood and anxiety disorders (Kendler, Kuhn & Prescott, 2004), it is important to understand the degree to which vulnerability and response to stress interact to influence decision making behaviour.

Based on prior results, we predicted that our healthy sample as a whole would demonstrate a bias towards avoiding disadvantageous decks at baseline (Bechara et al., 1994; Maia & McClelland, 2004), and that this bias would reverse under threat of shock (Preston et al., 2007; Van den Bos, Harteveld & Stoop, 2009). In our key exploratory analysis we also examined the interaction between stress and mood/anxiety symptoms on task performance. Based on epidemiological evidence for increased pathological mood/anxiety symptoms following stress in vulnerable individuals (Kendler, Kuhn & Prescott, 2004) and prior reports of greater disadvantageous deck selection in individuals with anxious and depressive symptomatology (Miu, Heilman & Houser, 2008; Must et al., 2013), we predicted that individuals with high trait vulnerability would exhibit maladaptive responses to stress, resulting in a reduction in the bias towards selecting advantageous decks.

Methods

Participants

Forty-seven participants completed the experiment (31 female: 16 male; mean age = 22.8, s.d. = 4.23). Ethical approval was obtained from the UCL Research Ethics Committee (1764/001). Participants were recruited via responses to an advertisement through the UCL Institute of Cognitive Neuroscience Subject Database and provided written informed consent. All subjects completed a screening form in which they reported that they were healthy with no history of psychiatric, neurological or cardiovascular diagnosis.

Procedure

Stress was induced by unpredictable electrical shocks delivered using a Digitimer DS7A Constant Current Stimulator (Digitimer Ltd., Welwyn Garden City, UK), with an electrode secured to the wrist. A shock work-up procedure standardised the level of shock for each individual. During the threat block the screen was red and displayed a message “YOU ARE NOW AT RISK OF SHOCK” whereas in the safe block the screen was blue with the message “YOU ARE NOW SAFE FROM SHOCK”. Participants were informed:

• You will alternate between safe conditions, where you will receive NO shocks, and threat conditions, where you might receive a shock at ANY time. THREAT conditions are indicated by a RED background, SAFE conditions are indicated by a BLUE background

• The shock is unpredictable and unrelated to your task performance but may come at any time during the threat condition.

At the end of each block, participants indicated how anxious they had felt during each of the threat and safe conditions on a scale from 1 (“not at all”) to 10 (“very much so”). This manipulation check has been used in numerous studies (see Robinson et al., 2013 for a review). Participants also provided self-report measures of depression (Beck Depression Inventory; BDI) (Beck & Steer, 1987) and trait anxiety (State Trait Anxiety Inventory: STAI) (Spielberger, Gorsuch & Lushene, 1970) at the end of the session.

Iowa Gambling Task

We adopted a forced-choice version of the Iowa Gambling task (Fig. 1) which enabled us to explore choice behaviour for all conditions (Cauffman et al., 2010) (Table 1). On each trial, one of four decks of cards was highlighted in a pseudo-random order (resulting in a minimum of 20 opportunities to reject each deck) by a yellow border, and the participant had to choose whether to play or pass that deck. If they chose to play, the participant was shown a monetary outcome (win, loss or no change), and a running monetary total displayed on-screen changed accordingly. If they passed, the next deck was selected (with no change in the monetary total). If no response was made within 4 s the card was passed automatically. The probabilities and magnitudes of win and loss outcomes varied between decks, such that two of the decks provided a net monetary gain on average if played repeatedly (advantageous decks; Table 1), while the other two decks provided a net loss (disadvantageous decks; Table 1). Participants played the game twice: once under threat of shock and once while safe (order counterbalanced) starting with new decks and a hypothetical $2,000 total in each of these two blocks. Each block consisted of 120 trials. Two shocks were delivered during the threat block (after the 13th and 58th trials). Participants were shown illustrated task instructions and completed 12 practice trials prior to their first block of the task.

Figure 1 Task schematic.

Subjects are given the option to play or pass the deck highlighted in yellow. Outcomes are added or subtracted from the running total displayed at the bottom of the screen. Subjects completed one baseline block whilst safe from shock and one stress block at risk of shock (order counterbalanced).

Table 1 Parameters for the possible monetary outcomes associated with each deck, adapted from Cauffman et al. (2010).

The net outcomes were equivalent to those in the original Iowa Gambling Task (Bechara et al., 1994).

	DECK	
	A	B	C	D	
Range	−$250–$100	−$1150–$100	−$25–$50	−$200–$50	
Probability of gain	50%	90%	50%	90%	
Probability of loss	50%	10%	25%	10%	
Probability of zero payoff	0%	0%	25%	0%	
Expected value (average
outcome over repeated choices)	−$25	−$25	$18.75	$25	

Analysis

Choice behaviour and reaction times were analysed using repeated-measures general-linear models (with between and within-subjects factors) in SPSS version 22 (IBM Corp, Armonk, NY). Data from the two advantageous and the two disadvantageous decks were pooled prior to analysis resulting in two choice variables. Choice behaviour (proportion of cards accepted) was then analysed in 3-way deck (advantageous, disadvantageous) × stress (threat, safe) × symptom (depression/trait anxiety as a continuous variable, in separate analyses) ANCOVAs. Post-hoc analyses were performed using Pearson’s r correlations between symptom measures and a measure of the impact of threat on choice (threat minus safe). Reaction time data were analysed in four-way deck (advantageous, disadvantageous) × stress (threat, safe) × decision (play, pass) × symptom (depression/trait anxiety as a continuous measure) ANCOVAs. Note that only N = 42 participants could be included in the reaction-time analysis as 5 never passed one or more of the decks and thus had incomplete reaction time data. Trait anxiety (mean = 41, a range = 21–64, standard deviation = 11) was normally distributed (Shapiro–Wilk test, p = 0.28), but BDI (mean = 6, range = 0–26, standard deviation = 6) was skewed towards lower values (Shapiro–Wilk test, p < 0.001) so a square root transformation was applied prior to analysis. For all analyses, P = 0.05 was considered significant. Based on the meta-analytic effect size of −0.58 for the Iowa Gambling Task (Mukherjee & Kable, 2014), t-test analysis in our sample (N = 41) has an ∼84% power to detect an effect size of alpha 0.01 (two-tailed). All data can be downloaded from figshare: http://dx.doi.org/10.6084/m9.figshare.1257693.

Results

Individual differences and manipulation check

There was a strong correlation between depression and trait anxiety measures (r(47) = 0.74, p < 0.0001).

Subjects rated themselves as significantly more anxious (F(1, 46) = 85.3, p < 0.0001, η2 = 0.65) during the threat condition (mean = 4.7/10, standard deviation = 2.3) than the safe condition (mean = 1.8/10, standard deviation = 1.2). This was not influenced by trait anxiety (F(1, 45) = 0.004, p = 0.95, η2 = 0.00009) or depression (F(1, 45) = 0.42, p = 0.52, η2 = 0.009) and there was no impact of order (F(1, 45) = 0.3, p = 0.6 η2 = 0.007. There was no impact of manipulation order on trait anxiety (F(1, 45) = 0.2, p = 0.7) or BDI (F(1, 45) = 0.3, p = 0.6).

Choice behaviour

We found a significant main effect of deck type (greater selection of advantageous decks: F(1, 46) = 36.2, p < 0.001, η2 = 0.44) but no stress by deck interaction (F(1, 46) = 0.43, p = 0.51, η2 = 0.009; Table 2). However, including trait anxiety in the model revealed a significant stress × deck × trait anxiety interaction (F(1, 45) = 6.4, p = 0.015, η2 = 0.13) which was driven by a significant correlation between trait anxiety and the stress-triggered propensity to play disadvantageous decks (r(47) = 0.336, p = 0.021) but not advantageous decks (r(47) = −0.065, p = 0.66; significant difference between correlations: Steiger’s Z = 1.96, p = 0.05; correlation with a compound advantageous vs. disadvantageous deck variable: r = −0.35, p = 0.015). In other words, we observed (along a continuum) stress-potentiated harm-avoidance in individuals with no/low anxiety symptoms, but the opposite pattern—stress-potentiated risk-seeking—in individuals with moderate anxiety symptomatology (Fig. 2). This was seen in the absence of a trait anxiety × deck interaction (F(1, 45) = 1.1, p = 0.31).

Figure 2 The effect of stress on participants’ propensity to play cards on threat blocks (proportion of cards accepted under threat minus proportion accepted under safe) varies significantly with anxiety and depression symptoms, for disadvantageous but not advantageous decks.

NB Raw depression scores are depicted but data were transformed with a square-root function for statistical analysis.

Table 2 Behavioural data for good and bad decks in threat and safe conditions across all subjects.

	Safe	Threat	
	Bad	Good	Bad	Good	
Choice	0.656 (0.032)	0.83 (0.019)	0.649 (0.03)	0.843 (0.019)	
	Play	Pass	Play	Pass	Play	Pass	Play	Pass	
RT	0.793 (0.039)	0.915 (0.041)	0.779 (0.041)	0.907 (0.052)	0.798 (0.039)	0.902 (0.041)	0.805 (0.044)	0.878 (0.058)	
Notes.

RT, reaction time in seconds (standard error of the mean); Choice, proportion of cards chosen from that deck.

Substituting trait anxiety for BDI in the model revealed a similar pattern of results. Critically, the significant stress × deck × depression interaction (F(1, 45) = 8.9, p = 0.005, η2 = 0.17) was also driven by a significant correlation between depression scores and threat-potentiated task performance for disadvantageous decks (r(47) = 0.415, p = 0.004) but not advantageous decks (r(47) = −0.012, p = 0.94; significant difference between correlations: Steiger’s Z = −2.1, p = 0.03; correlation with a compound advantageous vs. disadvantageous deck variable: r = −0.41, p = 0.005; Fig. 2). Again, no depression × deck interaction was seen (F(1, 45) = 1.3, p = 0.27).

Including both trait anxiety and depression scores in the same model resulted in the symptom × diagnosis × deck interactions becoming non-significant (depression, p < 0.12; anxiety, p < 0.57) indicating that the two scales account for the same variance in this sample.

There was a three-way manipulation order × stress × deck interaction (F(1, 45) = 12, p = 0.001, partial η2 = 0.2) as subjects were more likely to avoid the disadvantageous decks on their second block irrespective of condition. However, this learning effect was orthogonal to our effects of interest: including order had little impact on the results (trait; p = 0.012, η2 = 0.14; BDI p = 0.004, η2 = 0.17), and neither did including order in partial correlations between symptoms and threat-potentiated task performance for disadvantageous decks (trait; r = 0.344, p = 0.019; BDI r = 0.419, p = 0.004).

Reaction times

Subjects were faster to play than to pass cards (F(1, 41) = 26, p < 0.001, η2 = 0.39) but there was no main effect of stress (F(1, 41) = 0.008, p = 0.93, η2 < 0.001) deck (F(1, 41) = 0.33, p = 0.56, η2 = 0.008) or stress × deck interaction (F(1, 41) = 0.004, p = 0.95, η2 < 0.001; Table 2). Including symptoms in the model revealed no stress × deck × trait anxiety interaction (F(1, 40) = 0.256 p = 0.62, η2 = 0.006) or stress × deck × depression interaction (F(1, 40) = 0.02, p = 0.89, η2 < 0.001).

Post-hoc exploration of deck acceptance rates

Some participants (N = 5) failed to reject one of the four decks whenever it was presented. On the suggestion of one of the reviewers, to explore this further, we divided the sample into individuals who accepted any given deck more than 90% of the time (N = 23), and those who did not (N = 24). This ‘high acceptance’ rate (which was almost entirely driven by advantageous decks) showed an interaction with symptoms. Those who accepted any deck greater than 90% of the time showed significantly lower trait anxiety (F(1, 45) = 6.7, p = 0.012) and depression scores (F(1, 45) = 9.5, p = 0.003).

Discussion

The current study demonstrates an interaction between sub-clinical mood/anxiety symptoms and stress-responses on decision-making behaviour. Specifically, we found opposite stress-responses in those with low—versus those with moderate—anxiety and depression symptoms. On the one hand, individuals with low depressive symptoms displayed a pattern of stress-potentiated harm-avoidance, whilst those with moderate symptoms show an opposing pattern of stress-potentiated risk-seeking.

We replicated the frequently reported pattern of harm avoidance on the IGT across our sample as a whole (Maia & McClelland, 2004). Specifically, across all individuals, we saw a pattern of increased selection of advantageous over disadvantageous decks. We did not see, however, the predicted increase in selection of disadvantageous decks under stress across the whole sample. A close reading of prior findings indicates that this is likely a replication (Preston et al., 2007; Van den Bos, Harteveld & Stoop, 2009). The present study nevertheless extends our understanding of the role of stress to show that, in individuals with low depression or anxiety symptomatology, a baseline harm avoidant strategy (i.e., increased selection of advantageous vs. disadvantageous decks) is increased by stress. That is to say, low symptomatic individuals are even more likely to avoid risky decks under the stress condition relative to their individual baseline. This relative harm avoidant behaviour under stress may therefore be adaptive. Specifically, in conditions of threat, it may be wise to seek to minimise further loss. On the IGT this leads to fewer negative outcomes which, in a naturalistic setting, might mitigate the negative impacts of stress. One possibility, therefore, is that this pattern of stress-induced harm-avoidance and reduced anxious/depressive symptomatology is causally linked. Those who respond in a relatively more harm-avoidant manner to stress might be better at regulating their environmental exposure to sources of stress, and ultimately reduce their vulnerability to mood and anxiety disorders.

By contrast, in individuals with moderate symptomatology, the opposite pattern was evident. Stress provoked a pattern of relatively increased risk-seeking as indexed by greater selection of disadvantageous decks under stress. This pattern has been observed in individuals with high mood and anxiety disorder symptomatology (Must et al., 2013; Walteros et al., 2011; Miu, Heilman & Houser, 2008), and in response to social stress in some individuals (Walteros et al., 2011; Werner, Duschek & Schandry, 2009; Preston et al., 2007; Van den Bos, Harteveld & Stoop, 2009). Here we extend and integrate these findings to show that this effect is specific to stress-response in symptomatic individuals. On the one hand, this response can be seen as maladaptive. Subjects appear to be seeking further risk, which on average will lead to increased losses on this task, potentially increasing stress exposure in a vicious cycle. On the other hand this might simply reflect an alternative, albeit more risky, strategy. The disadvantageous decks do actually demonstrate larger occasional gains (as well as losses; Table 1). So, if a player is lucky, they could potentially more rapidly improve their overall gains, especially if they only occasionally sample from the risky decks. However, in the long run this strategy will not pay off on the present task and will lead to increased overall losses. This alternative strategy may therefore drive elevated susceptibility to pathological symptoms; the more likely an individual is to adopt relatively more risky strategies, the more vulnerable they are to negative outcomes, associated negative mood states, and hence mood and anxiety disorders. Indeed, there is some naturalistic evidence for such a mechanism. Some individuals with high levels of social anxiety demonstrate high levels of risk-taking behaviour on questionnaire measures (Kashdan, Collins & Elhai, 2006; Kashdan, Elhai & Breen, 2008; Kashdan & McKnight, 2010). This may, in turn, explain some of the discrepancies across prior studies. Trait anxiety, for instance, has been shown to be associated with harm-avoidant behaviour on the Iowa Gambling Task in some studies (Mueller et al., 2010) and an opposite pattern of risk-seeking in other studies (Miu, Heilman & Houser, 2008). Taking into consideration a given subject’s current stress levels may go some way towards explaining this apparent variability.

It is beyond the scope of the present paper to explore the neurochemical basis of this effect, but prior work has shown that the anxiogenic effect of threat of shock is thought to be modulated in part by serotonin and CRH in the amygdala and bed nucleus of the stria terminalis (Davis et al., 2010; Robinson et al., 2012b). It should also be noted that in the present study we were interested in the effect of our manipulation on cognitive measures and as such we did not obtain psychophysiological measures from our participants. Nevertheless, extensive prior work has shown that the threat of shock manipulation reliably and reversibly increases startle responding across humans and animal models (for reviews see Grillon, 2008; Grillon et al., 1991; Davis et al., 2010; Robinson et al., 2013). Finally, our paradigm is designed as a translational anxiety/stress induction, but it might be interesting to explore specificity via comparison with other manipulations designed to elicit, for instance, anger or happiness, or with other anxiety manipulations such as oxygen deprivation.

This study raises a number of questions. First, is this effect on disadvantageous decks simply an epiphenomenon of increased current mood disorder symptomatology, rather than underlying vulnerability? From our data we cannot determine the causal relationship between symptoms and stress-induced behavioural change. This question might be addressed through the use of longitudinal designs that follow-up with subjects and determine the relationship between stress-induced behavioural change and subsequent disorder onset. Secondly, why do we see the same effect in trait anxiety and depression scores? Given that they largely account for the same variance (i.e., including both in the same model leads to both interactions becoming non-significant) this may be because both measures tap into the same broad ‘negative affect’ construct. The significant positive correlation between the measures in this sample provides further support for this as, indeed, does the observation that depression and anxiety disorders are highly co-morbid (Kessler et al., 2005). Future work should seek to resolve whether depression and anxiety symptoms actually represent separate constructs at a mechanistic level. Finally, our post-hoc analysis of ‘high deck acceptors’ (i.e., those who selected any deck more than 90% of the time) indicates that individuals with lower symptoms might have a greater propensity to explore possibly advantageous choices in the face of uncertainty (i.e., accept over 90% of uncertain gambles on advantageous decks). This is, of course, speculative, but future work may wish to explore whether this has any functional significance: does it promote resilience to symptoms in some way?

Additional Information and Declarations

Competing Interests

Author Contributions

Human Ethics

Data Deposition

The authors declare there are no competing interests.

Oliver J. Robinson and Rebecca L. Bond conceived and designed the experiments, performed the experiments, analyzed the data, contributed reagents/materials/analysis tools, wrote the paper, prepared figures and/or tables, reviewed drafts of the paper.

Jonathan P. Roiser conceived and designed the experiments, analyzed the data, contributed reagents/materials/analysis tools, wrote the paper, prepared figures and/or tables, reviewed drafts of the paper.

The following information was supplied relating to ethical approvals (i.e., approving body and any reference numbers):

UCL Research Ethics Committee: 1764/001.

The following information was supplied regarding the deposition of related data:

Figshare: http://dx.doi.org/10.6084/m9.figshare.1257693.

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
