# Peer review of "The impact of stress on financial decision-making varies as a function of depression and anxiety symptoms"

_PeerJ, doi:10.7717/peerj.770_

## Round 0.1 · original submission · Major Revisions

The manuscript has received major criticisms but we can reconsider it in case the authors accept to revise it in accordance to reviewers' comments.

Reviewer 1 ·

Basic reporting

Review of the study: The impact of stress on financial decision-making varies as a function of depression and anxiety symptoms
By Oliver J Robinson, Rebecca Bond, Jonathan P Roiser

This study examines a topic which is very up-to-date and important both for its practical implications (understanding of the psycho-pathological states) and for its theoretical implications. As regards the latter aspect, it is sufficient to mention only that there are very few studies (about a dozen) that have investigated the relationship between stress and risk-taking behavior and even less (only two) are the studies that have done this by using a task such as that used by the authors, i.e. the Iowa Gambling Task.

It is with great regret that I write these lines in which, as you'll see I delineate a series of problems of the study which undermine the validity and resize the impact within the search field. The beautiful results found by the authors are not demonstrable, because they are very likely the result of a series of methodological errors and explanations are therefore not supported. I am very sorry because the domain really needs more studies. Stress is a topic that becomes every day more important and we do not know almost nothing about how stress impacts the individual propensity to take risks.

I list here the problems:

1. The main problem with the study is a methodological problem. The authors state that they induced stress in the subjects using the paradigm of the electric shock. Unfortunately the authors do not have any proof of the fact that this really happened, that is, that the stress was really induced. Authors do not have a measure of "manipulation check", if not a response to a single question "how did you feel anxious", replicated twice after each block (the stress block and the safe block). It is really too little. Among other things, the question is only one, it is not even a scale (series of item) and it is not even validated. A simple measure of heartrate would have solved the problem.
2. Connected to the previous problem is the fact that the same subject was first placed in a condition of stress (probability of shock) and then placed in a neutral condition (zero probability of electric shock). The within-subjects design is not correct if you do not leave a delay between sessions of at least one week. How can you avoid that the stress induced by the previous block is not passed on to the next? It is impossible. I cannot say to cortisol in the bloodstream to return from where he came from because we now are in the safe block. The mind has limits on the physic and stress hormones take a while to stabilize!
3. Always connected to this problem is the fact that the electric shock, used in the way that was used in this study, it is not a validated paradigm. The classic fear conditioning task uses other paradigms (two squares alternated) (Olsson & Phelps, 2007). The task should be first validated with measures of heart rate, skin conductance, cortisol, to say that it induces stress.
4. Another problem is the type of stress induced. But this is not a problem of the study but of all the field of investigation that uses stress inductions. No one makes case to the type of emotion induced along with stress. Instead we know very well that if I induce fear (negative emotion) this is followed by an increase in risk aversion and if I induce anger (negative emotion) this is followed by an increase in risk seeking (Kugler, Connolly, & Ordo, 2012; Lemer, Lerner, & Keltner, 2001). If I induce negative stress with an electric shock, very probably I induce even an emotion, or not? AND if you do induce an emotion, which one? Very nice in this regard are studies on risk-taking with hypoxia in which the stress manipulation is only physical and unconscious and not paired with emotions (Pighin, Bonini, Savadori, Hadjichristidis, & Schena, 2014; Pighin & Schena, 2012).
5. It is not indicated if the measure of trait anxiety was always performed after the test to the Iowa Gambling Task. If so, I fear that the self-report measure can be soiled from stress. Those in which the self-report measure was collected after the block with electric shocks should give higher responses of anxiety than those for which the measure was collected after the safe-blocks. In other words, the order of the blocks should count on trait-anxiety. Why are no statistics reported to discredit this? In fact the questions are of the type "I feel calm" (almost never- almost always) and are easily misunderstood to answer "how do you feel now", especially after an induction of stress. The measures were to be taken before, and perhaps even outside the context of the study, for example, two-three months before, on the occasion of another study.
6. It is not indicated what is said to the subjects about the probability of receiving an electric shock: was it said what this probability was?
7. It is not indicated if the order (first condition stress and then neutral condition) has influenced the results.
8. It is not indicated if the subjects came in the lab to do a session of familiarization before the real test. It is well known that the first time that the subjects are in the laboratory they are inevitably in a stressful condition for the newness of the place and task. Has this been controlled for? Unfortunately there is no way to know because there are no measures of stress. Furthermore, the measures to the question "you're anxious" to the first block of trials don't say anything because they are soiled from stress manipulation.
9. It is not indicated whether subjects did some exercise trials on the task.
10. Even if the study was methodologically valid, the authors should find a good argument of why their results do not replicate those already found in a previous study in which stress was manipulated and it’s effect measured on the behavior in the IGT (Preston, Buchanan, Stansfield, & Bechara, 2007; van den Bos et al., 2009)?
11. The "newness" of this study (which I am afraid it is only a methodological error) is very little because it is only a replication of previous results (Preston et al. , 2007) with the addition that the results replicate only if you control for the error induced by the variance in trait-anxiety of the subjects.
12. The studies on the effect of stress on the IGT find strong gender differences (Preston et al. , 2007; van den Bos, Harteveld, & Stoop, 2009). Why in the present study authors do not find such differences?
13. The two sub-samples (men vs. women) are not homogeneous in number and this suggests that the authors did not have in mind to study the effect of trait anxiety on stress because they would have had to consider the variable sex in face planning as in studies where you measure the trait-anxiety on the IGT (Miu et al., 2008).
14. A big error lies in the interpretation of the data. The authors can say they have found that those who have higher trait anxiety, choose more from disadvantageous decks when they are under stress rather than when they are in the neutral condition, and that's it. They can not talk about risk seeking or risk aversion or harm avoidance or adaptive behavior or not adaptive. It is not known what the correct behavior (how many times theoretically would be advantageous to choose the disadvantageous deck compared to the times that this was presented?). There is no normative baseline on which to qualify the behavior of the subjects in harm avoidant or risk seeking . All I can say is relative. You can tell that subjects turned more cards from the disadvantageous decks when they were under stress, but you can not say that under stress their behavior was adaptive, perhaps because even under stress they turned too many disadvantageous cards. It was more adaptive in relative terms, but not in absolute terms.
15. The type of modified version of the IGT used is unclear. Based on what the computer offered the deck to be turned? The choice of the deck I make inevitably depends on which deck the computer offered me. How this factor was controlled in terms of statistical analysis?
16. The authors do not explain what type of statistical analysis they conducted. Since the dependent measure is a dichotomous variable (good decks vs bad decks) I suppose they used a logit model. Since the design is within-subjects then the responses of the individuals are dependent upon each other. Did they entered subjects as a random intercept (mixed models)?

There are other flaws but I will stop here. I really encourage the authors to re-run the study in a more clean methodological way (not within but between, if possible) because I do believe they are on a good and promising track.


Kugler, T., Connolly, T., & Ordo, L. D. (2012). Emotion , Decision , and Risk : Betting on Gambles versus Betting on People, 134(December 2010), 123–134. doi:10.1002/bdm
Lemer, J. S., Lerner, J. S., & Keltner, D. (2001). Fear, anger, and risk. Journal of Personality and Social Psychology, 81(1), 146–59. Retrieved from http://www.ncbi.nlm.nih.gov/pubmed/23708477
Miu, A. C., Heilman, R. M., & Houser, D. (2008). Anxiety impairs decision-making: Psychophysiological evidence from an Iowa Gambling Task. Biological Psychology, 77, 353–358. doi:10.1016/j.biopsycho.2007.11.010
Olsson, A., & Phelps, E. A. (2007). Social learning of fear. Nature Neuroscience, 10(9), 1095–1102.
Pighin, S., Bonini, N., Savadori, L., Hadjichristidis, C., & Schena, F. (2014). Loss aversion and hypoxia: less loss aversion in oxygen-depleted environment. Stress (Amsterdam, Netherlands), 17, 204–10. doi:10.3109/10253890.2014.891103
Pighin, S., & Schena, F. (2012). Decision making under hypoxia : Oxygen depletion increases risk seeking for losses but not for gains, 7(4), 472–477.
Preston, S. D., Buchanan, T. W., Stansfield, R. B., & Bechara, a. (2007). Effects of anticipatory stress on decision making in a gambling task. Behavioral Neuroscience, 121(2), 257–63. doi:10.1037/0735-7044.121.2.257
Van den Bos, R., Harteveld, M., & Stoop, H. (2009). Stress and decision-making in humans: Performance is related to cortisol reactivity, albeit differently in men and women. Psychoneuroendocrinology, 34, 1449–1458. doi:10.1016/j.psyneuen.2009.04.016

Experimental design

see above

Validity of the findings

see above

Additional comments

see above

·

Basic reporting

This is a very welll written and clear manuscript. It frames the current paper in terms of prior research very nicely. I think that the topic of the paper is appropriate for publication.

Experimental design

The experimental design appears to be appropriate.

I have a couple of recommendations about the description of the study methods and the reporting of the results that the authors may want to consider:

1. I think the authors need to provide more detail about the probabilities and magnitudes of the wins and losses associated with each of the 4 card decks. At present they just report that the decks are advatanagous or disadvantagous, with the disadvantagous decks containing more high magnitude wins. I think readers would benefit from a more complete description of what the probabilities and magnitudes actually were.
2. On reading the results it seemed to me that the purest measure of behaviour (i.e. as risk seeking or harm avoiding) would be provided by the difference score, calculcated for each participant, of the tendency to select advantagous - disadvantagous decks. Figure 2 has split these into two different graphs, which , although giving a finer grained depiction of the results, obscures this key behavioural measure. I wonder if it might be useful to incorporate a graph showing this result into figure 2. If it was felt that this would clutter the figure too much, then I think it may be possible to drop either the depression or anxiety results-- as the authors discuss, these two measures seem to be capturing the same underlying tendency which can't be separated with the current data, so I am not sure how useful it is to display what is essentially the same result twice.
3. It was interesting that a number of participants did not sample all of the decks during the task (5 in total). This seems like a disadvantgous strategy. I wonder if there is anyway to invesitgate whether the tendency to limit the search across decks varies as a function of threat of shock and/or trait anxiety depression? If possible, this may provide some insight as to the cause of the disadvantguous resposne to stress in the high anxious/depressive participants (e.g. perhaps they respond to stress by reducing exploration). As only 5 people did not sample certain decks at all, it may be that there are too few data points to look at this question in a binary fashion, although it may be that the tendency to adequatley sample the decks could be estimated with a more continuous outcome (e.g. minimum number of times one of the 4 decks was sampled, or even the decision temperature of a formal model). This suggestion is fairly speculative, so I would be happy to be informed by the authors if it is not feasible.

Validity of the findings

I think the results are generally fine. I only have two fairly minor points about the discussion section:

1. I think it should be highlighted at the start of the discussion that the clear hypothesis of the study-- that the threat of shock would lead to an increase in disadvantagous choice-- was not confirmed. This is an interesting result in itself. It is fine to go on to discuss, as the authors do, the qualification of this result by the effect of anxiety and depression, but the basic nul effect needs to be a bit more prominent.

2. The section on the application of this task in clinical settings:
"Moreover,these findings raise the possibility of using this stress manipulation as a screen for risk for common mental health problems. Specifically, it might be possible to develop an emotional ‘stress-­‐test’ to probe individual differences in stress-­‐responding in a controlled laboratory environment prior to the onset of a disorder. Future work wil explore this possibility."
Is really pretty speculative given the extremely early stage of this research (i.e. no main effect of stress, with an interaction between the stressor and continuous measures of negative affectivity in a non-clincial sample). I really think that either, the prospect of using this in a clincial setting should be highlighted to be extremely remote, or this paragraph should be removed.

Additional comments

I think this is a very nice study and with a few, realtively minor changes I would be happy to support its publication

---

## Round 0.2 · accepted · Accept

Congratulations on this acceptance.

·

Basic reporting

I think that the revised manuscript reports the study and findings well. I have no further comments

Experimental design

This is a clearly defined and well conceived study. I have nothing further to add.

Validity of the findings

I think the analysis of the data is appropriate and have nothing further to suggest

Additional comments

This is an interesting article. The authors have responded well to all of my concerns and I see no reason to ask for any further alterations. I would recommend that the paper is accepted for publication.